

# Relationships among barodontalgia prevalence, altitude, stress, dental care frequency, and barodontalgia awareness: a survey of Turkish pilots

Celalettin Topbaş[1], Dursun Ali Şirin[1], Hilal Gezeravcı[1], Fatih Özçelik[2], Yelda Erdem Hepşenoğlu[3] and Şeyda Erşahan[3]

[1] Faculty of Dentistry, Department of Endodontics, University of Health Sciences, Istanbul, Üsküdar, Türkiye
[2] Hamidiye Etfal Training and Research Hospital, Department of Medical Biochemistry, University of Health Sciences, Istanbul, Sisli, Türkiye
[3] Faculty of Dentistry, Department of Endodontics, Istanbul Medipol University, Istanbul, Esenler, Türkiye

Corresponding author
Celalettin Topbaş,
celalettin.topbas@sbu.edu.tr

## ABSTRACT

**Background**. Gas expansion in body cavities due to pressure changes at high altitudes can cause barodontalgia. This condition may compromise flight safety.

**Aim**. To investigate relationships among barodontalgia awareness, dental visit frequency, and barodontalgia prevalence in civilian and military pilots operating at high altitudes.

**Materials and Methods**. Civilian pilots from Turkish Airlines and military pilots from the Turkish Air Force, flying between November 2022 and January 2023, participated in this study. A 20-question survey was administered to 750 pilots, covering topics such as barodontalgia awareness, dental visit frequency, breaks after dental treatments, in-flight pain, and pain type and severity. The voluntary surveys were distributed by email.

**Results**. Of the 750 pilots, 526 completed the survey; 61% were aware of barodontalgia, and 81% of pilots who had experienced it reported pain at altitudes <2000 feet. The study revealed higher barodontalgia awareness among pilots who had experienced it, with the highest prevalence among jet pilots. Pilots with barodontalgia also showed a higher frequency of dental visits ($p < 0.001$). Additionally, this group reported more frequent interruption of flight due to dental treatment (IFDT), more problems experienced in flights after treatment (PFAT), and higher instances of bruxism or teeth clenching during flight, suggesting stress and anxiety ($p < 0.05$).

**Conclusions**. Barodontalgia, a type of pain linked to stress, significantly impacts pilot performance, and can threaten flight safety, even at lower altitudes. Thus, there is a need to educate pilots about stress management, barodontalgia awareness, and the importance of regular dental check-ups.

## INTRODUCTION

Barodontalgia, a toothache induced by altitude and pressure changes, affects flight crews, airline passengers, and pilots. Increasing altitude during flight causes ambient pressure to

**Table 1  Classification of Barodontalgia.**

| Class | Pathology | Features | Flight position |
|-------|-----------|----------|-----------------|
| I | Irreversible pulpitis | Sharp, momentary, transient pain | Ascent |
| II | Reversible pulpitis | Dull, throbbing pain | Ascent |
| III | Necrotic pulp | Dull, throbbing pain | Descent |
| IV | Periapical pathology | Severe, persistent pain | Both (Ascent and Descent) |

drop, reducing pressure on the body and causing gas volumes in the body to expand, as described in Boyle's law (*Kieser, 1997*). Conversely, deep sea diving increases pressure on the body, leading to decreased gas volume. Pain arises when gases in closed areas of the body, such as the pulp and sinuses, cannot adjust to these pressure changes (*Kollmann, 1993*). This condition often manifests in the lungs, middle ear (barotitis media), or maxillary sinus (barosinusitis) (*Robichaud & McNally, 2005*; *Gonzalez Santiago, Martinez-Sahuquillo Marquez & Bullon-Fernandez, 2004*; *Zadik, 2006*). Extreme pressure differences may lead to serious complications, including fragile oral mucosa and loss of consciousness, which carry flight safety risks (*Al-Hajri & Ebtissam, 2006*; *Zadik, Chapnick & Goldstein, 2007*; *Sipahi et al., 2007*). Therefore, it is important to ensure accurate differential diagnosis of toothaches triggered by barometric changes.

Barodontalgia is not a pathological condition itself, but a symptom of inflammation related to a pre-existing subclinical oral disease. Various oral pathologies, such as faulty tooth restorations, caries not reaching the pulp (29.2%), necrotic pulp or periapical inflammation (27.8%), vital pulp pathology (13.9%), recent dental treatment (11.1%), and barosinusitis (9.7%), have been identified as etiological factors in barodontalgia (*Kollmann, 1993*; *Zadik, Chapnick & Goldstein, 2007*; *Ferjentsik & Aker, 1982*; *Jamil, Reilly & Cooper, 2024*). Barodontalgia has been reported at altitudes of 600–1,500 m in flight and depths of 10–15 m in diving, with prevalences of 0.26–8% among flight crews (*Hamilton-Farrell & Bhattacharyya, 2004*). Ferjentsik and Aker's diagnostic classification for barodontalgia includes underlying causes (reversible, irreversible, necrotic pulpitis, and periapical pathology) and clinical symptoms (sharp, momentary, transient, dull, persistent, severe, and throbbing pain) (*Ferjentsik & Aker, 1982*; *Zadik, 2009a*) (Table 1).

Barodontalgia prevention and treatment involve awareness and timely treatment of dental issues. For high-risk groups such as flight crews and divers, regular dental examinations (every 6 months) and radiological assessments with panoramic and periapical X-rays are recommended (*Al-Hajri & Ebtissam, 2006*). Monitoring should focus on faulty restorations, deep caries, periapical lesions, bruxism, and attrition (*Robichaud & McNally, 2005*; *Zadik & Einy, 2006*). Temporary flight restrictions after dental treatments and surgical procedures are effective in preventing barodontalgia (*Gonzalez Santiago, Martinez-Sahuquillo Marquez & Bullon-Fernandez, 2004*; *Lurie et al., 2007*). This study explored relationships among barodontalgia awareness, dental visit frequency (DVF), and barodontalgia prevalence in civilian and military pilots operating at high altitudes.

## METHODS

### Ethical board approval

This randomized, parallel, single-blind clinical trial was approved by the Institutional Ethics Committee (15.11.2022-13351). The study was evaluated at the meeting of the University of Health Sciences Hamidiye Scientific Research Ethical Board.

### Sample size calculation and power analysis

Prior power analysis (PS Power and Sample Size Program, Version 3.1.2) was conducted using data from a recent study (*Daud et al., 2019*) that compared barodontalgia levels between commercial and military pilots. The minimum sample size required for this study, with $\alpha = 0.05$ and power $= 0.80$, was initially calculated as 14 subjects. However, considering that the impact size in the study was >1, adjustments were made to achieve the conventionally recommended impact size of 0.5; the minimum sample size was recalculated as 32.

### Experimental design

This study involved civilian pilots from Turkish Airlines and military pilots from the Turkish Air Force who flew between November 2022 and January 2023. This study used a simple random sampling technique in which each item included in the research had an equal probability of being selected. All pilots who were on the aviation social activity group list were invited to participate in the survey *via* e-mail. Questionnaires were emailed to 750 pilots, and participation was voluntary (Table 2). In total, 526 pilots responded to the questionnaire. Before completion of the questionnaire, each participant was required to read an informed consent form. Participants who agreed to participate in the study were asked to indicate their consent by selecting the statement, "I agree to participate in the study."

### Statistical analysis

Data collected were analyzed using IBM® SPSS® Statistics for Windows, version 25 (IBM Corp., Armonk, N.Y., USA). Study data were summarized in tables in accordance with descriptive (mean, standard deviation, median, minimum and maximum) and analytical research methodology. For comparisons between groups of categorical data [such as gender, jet/commercial pilot, knowledge of barodontalgia (KB), filling, root canal treatments (RCT), tooth extraction, implant, sub/supragingival scaling, interruption of flight due to dental treatment (IFDT), problems experienced in flights after treatment (PFAT), bruxism, clenching teeth during flight (CTDF) and pain in the jaw joint during clenching (PJDC)] answered as yes/no or yes/no, Chi-square test was used. Unpaired t test was used to compare the data of the groups that were found to be parametric (such as age), and the Mann–Whitney U test was used to compare the data that were found to be non-parametric, such as dental visit frequencies. The statistical significance level was accepted as $p < 0.05$.

**Table 2    Questions of questionnaire.**

**Questions asked to the pilots.**

- What is your gender?
- What is your age range?
- What field of aviation are you working in? (Civilian-Military)
- Have you ever heard about the concept of barodontalgia?
- How often do you visit the dentist?
- What treatment(s) were applied to you in those visits?
- Have you ever had a flight interruption because of a dental procedure?
- Did you encounter any issues on flights immediately following dental treatment?
- Do you habitually clench or grind your teeth?
- Do you clench your teeth during flights?
- Does your habit of clenching your teeth cause any pain in your jaw joint?
- Have you ever experienced a toothache while flying?
- Please rate the severity of the pain you experienced during the flight on a scale of 1 to 5.
- How would you describe the pain you experienced?
- How long did the pain last during the flight?
- At what stage of the flight did you experience the pain?
- At what altitude range did the pain occur?
- Did the pain dissipate immediately after the flight?
- Was a dental examination conducted following your complaint related to the flight?
- What was the cause of the pain determined upon examination?

## RESULTS

There was no statistically significant difference in sex between groups with and without barodontalgia ($p > 0.05$). However, the mean age was older in the group with barodontalgia ($41.4 \pm 7.5$, $p = 0.0002$) (Table 3). The number of fighter jet pilots was significantly greater in the barodontalgia group than in the group without barodontalgia ($p < 0.0001$). Regarding KB, a higher KB ratio was observed in the group with barodontalgia ($p < 0.0001$) (Table 3), highlighting the importance of awareness. The group with barodontalgia also had a higher DVF ($p < 0.0001$) (Fig. 1). Comparison of the treatments received showed that the rates of fillings, RCT, tooth extractions, implants, periodontal procedures, prosthetic, and orthodontic treatments were significantly higher in the barodontalgia group ($p < 0.05$).

The rates of IFDT and PFAT were considerably higher in the barodontalgia group ($p < 0.0001$). These findings suggest that costly and risky flights could be delayed or interrupted due to barodontalgia after dental treatment. Bruxism, associated with anxiety and/or stress, and teeth clenching during flight were more prevalent in the barodontalgia group ($p < 0.0001$). Additionally, the rate of CTDF and PJDC was higher in this group ($p < 0.0001$). These findings indicate that personal characteristics, stress, and anxiety contribute to barodontalgia. Stress-induced teeth clenching and/or grinding can lead to anatomical changes in teeth, reducing resistance to impacts and increasing susceptibility to fractures and bacterial infections. Thus, pilots exhibiting both barodontalgia and bruxism should receive psychological counseling.

**Table 3  Analysis of demographic data comparing pilot groups with and without barodontalgia.**

|  | All Pilots (A) | Pilot group without barodontalgia (B) | Pilot group with barodontalgia (C) | *p* values |
|---|---|---|---|---|
| **n** | 526 | 239 | 287 | – |
| **Age, year** | 40.3 ± 7.7 | 38.9 ± 7.8 | 41.4 ± 7.5 | [a]0.0002 |
| **Gender, F (%)** | 37(7%) | 18(7.5%) | 19(6.6%) | [c]0.4289 |
| **Jet/Commercial pilot, n** | 255(48%)/271 | 74(31%)/165 | 181(63%)/106 | [c]<0.0001 |
| **KB, n** | 321(61%) | 111(46%) | 210(73%) | [c]<0.0001 |
| **DVF, n** | 2.0(1.0-3.0) | 2.0(1.0-3.0) | 3.0(1.0-3.0) | [b]<0.0001 |
| **Filling, n** | 338(64%) | 127(53%) | 211(73%) | [c]<0.0001 |
| **RCT, n** | 253(48%) | 72(30%) | 181(63%) | [c]<0.0001 |
| **Tooth extraction, n** | 113(%21) | 27(11%) | 86(30%) | [c]0.0002 |
| **Implant, n** | 124(24%) | 31(13%) | 93(32%) | [c]0.0001 |
| **Sub/supragingival scaling** | 286(54%) | 157(66%) | 129(45%) | [c]<0.0001 |
| **Prosthetic** | 24(%4.6) | 0(0%) | 24(8%) | – |
| **Orthodontic** | 82(%15.6) | 22(9%) | 60(21%) | [c]0.0153 |
| **IFDT, n** | 205(39%) | 17(7%) | 188(65%) | [c]<0.0001 |
| **PFAT, n** | 177(34%) | 6(2.5%) | 171(60%) | [c]<0.0001 |
| **Bruxism, n** | 247(47%) | 53(22%) | 194(68%) | [c]<0.0001 |
| **CTDF, n** | 168(32%) | 17(%7) | 151(%53) | [c]<0.0001 |
| **PJDC, n** | 192(36%) | 16(7%) | 176(61%) | [c]<0.0001 |

**Notes.**

[a]Unpaired t test
[b]Mann–Whitney U Test
[c]Chi-square test

Mean ± standard deviation was used for parametric data and median (min-max) for nonparametric data. Dental procedures include filling, root canal treatment (RCT), tooth extraction, implant placement, sub/supragingival scaling, prosthetic fitting, and orthodontic treatment.

KB, Knowledge of barodontalgia; F, Female; DVF, Dental visit frequencies (in case of complaint: 1, once a year: 2, every six months: 3); IFDT, Interruption of flight due to dental treatment; PFAT, Presence of problems in flights immediately after dental treatment; Bruxism, Teeth clenching and/or grinding; CTDF, Clenching teeth during flight; PJDC, Pain in the jaw joint during clenching.

Pilots experiencing barodontalgia during flight rated their pain severity as ~4/5 (Table 4). Seventy-two (25%) of the pilots with barodontalgia reported that their symptoms vanished after flight (reversible pulpitis), indicating the persistence of symptoms in 75% (irreversible pulpitis or necrosis). This result suggests that barodontalgia is not temporary and can continue even after pressure normalization. Among the pilots with barodontalgia, 256 (89%) visited a dentist after flight, reflecting the magnitude of the issue. Questionnaire results showed that most barodontalgia cases involved throbbing pain (66%) and pain duration >5 min (89%). Most of affected pilots reported that pain mostly occurred during descent. Overall, 81% of pilots experienced barodontalgia at low altitudes (0–2,000 ft); caries, prior treatments (fillings, RCT), and bruxism were the most significant etiological factors (*Ferjentsik & Aker, 1982*).
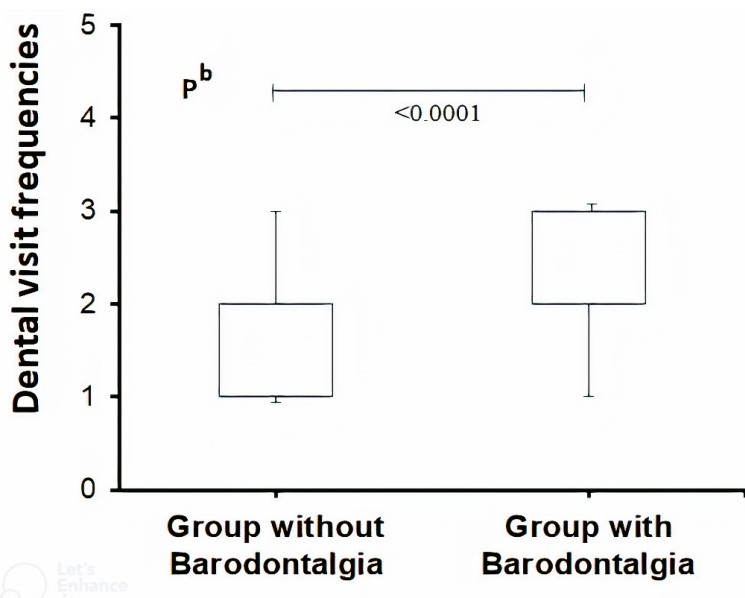

**Figure 1** Nonparametric box plot of dental visit frequencies (DVF) for different levels of frequency (complaint: 1, once a year; 2, every six months; 3, in pilot groups with and without barodontalgia). The DVF of the group with barodontalgia was significantly higher compared to the group without barodontalgia (Mann–Whitney U Test, non-parametric).

## DISCUSSION

This large-scale survey study with 526 participants aimed to assess DVF, barodontalgia prevalence, KB, IFDT, PFAT, and pain type and severity relative to altitude among military and civilian pilots. The participants were predominantly men (93%); only 7% were women. Civilian and commercial pilots represented 51.5% of the cohort, whereas military fighter jet pilots comprised 48.5%. The study revealed that 45% of the participants did not experience any in-flight pain, while 55% of pilots experienced barodontalgia. These findings are consistent with a Kuwaiti study reporting a similar barodontalgia rate (49.6%) (*Al-Hajri & Ebtissam, 2006*), but Indian and Brazilian pilots reported a lower rate (20.6% and 21.7%, respectively) (*Rai et al., 2010*; *Lipisk et al., 2023*). A Jordanian military pilot study showed a significantly lower prevalence of 10.49%, which differed from our findings (55% in total) (*Al-Khawalde et al., 2016*). Turkish and Israeli studies showed barodontalgia rates of 12% (*Sipahi et al., 2007*) and 8.7%, respectively (*Jamil, Reilly & Cooper, 2024*).

According to the results of our study, as DVF increased, KB also increased. According to these findings, pilots with poorer oral hygiene may have experienced more dental pain and visited the dentist more often, and their KB may have increased with more information provided by the dentist. Similarly, pilots with less DVF may have better oral hygiene. Therefore, as a result of fewer visits to the dentist, they may have received less information from the dentist about barodontalgia, and therefore their KBs may have been lower. The fact that 45% of the pilots had never experienced barodontalgia, while 29.5%, 17.4%, and 8% had experienced barodontalgia once, twice, and three times, respectively, could also be

**Table 4  Characteristics of barodontalgia during flight.**

| | | Pilot group with barodontalgia |
|---|---|---|
| n | | 287 |
| SB, degree | | 4.0 (0.0–5.0) |
| DBAF, n | | 72 (25%) |
| NDEFB, n | | 256 (89%) |
| Type of barodontalgia | Throbbing, n | 188 (66%) |
| | Tingling, n | 99 (34%) |
| Pain duration, n | A few seconds | 3 |
| | 1–5 min | 29 |
| | 6–30 min | 141 |
| | >30 min | 114 |
| TFIB, n | Take-off | 57 |
| | Landing | 253 |
| | Straight flight | 143 |
| | CTF | 12 |
| FABB, fit | 0–2000 | 233 (81%) |
| | 2001–5000 | 229 (80%) |
| | ≥5001 | 121 (42%) |
| Cause of barodontalgia, n | Not found | 20 |
| | Caries | 156 |
| | Previous treatments | 170 |
| | Bruxism | 99 |
| | Abscess | 8 |
| | Orthodontic | 9 |
| | Sinusitis and other | 9 |

**Notes.**

SB, Barodontalgia severity (none: 0, very mild: 1, mild: 2, moderate: 3, severe: 4, very severe: 5); DBAF, Disappearance of barodontalgia after flight; NDEFB, The number of dental exams after flight-related barodontalgia; TFIB, It is the time to the start of flight-induced barodontalgia; CTF, Continuously throughout the flight, the complaint of barodontalgia can occur; FABB, Flight altitude at which the complaint of barodontalgia begins; Old treatments, Old treatments include filling and/or root canal treatmen.

In TFIB, FABB, and barodontalgia causes, participants may choose more than one option.

attributed to their good oral hygiene and the types of pathologies that may be present in their mouths, the absence of a serious lesion, or the relatively small size of the lesion.

Regarding KB, 61% of the participants were aware of barodontalgia, but 39% were not. The frequency of dental visits varied among the pilots: 34.4% and 35.2% had regular check-ups every 6 months and annually, regarding, whereas 30.4% visited the dentist only when they experienced problems. Higher KB and DVF were observed in the group with barodontalgia. These findings suggest that the experience of flight-related toothache led to increased DVF and barodontalgia awareness. In contrast, pilots without such experiences had lower KB and DVF.

Barodontalgia, often triggered by barotraumas such as barosinusitis and barotitis media (*Jayasrikrupaa et al., 2020*), is not a pathological condition itself but a symptom of pre-existing conditions exacerbated by pressure changes during flight or diving. Underlying

causes can include faulty restorations, caries, pulpitis, pulp necrosis or gangrene, periapical lesions, various periodontal diseases, sinusitis, otitis media, temporomandibular joint disorders, and jawbone pathologies.

In the high-altitude simulation study by *Kollmann (1993)* the most common cause of barodontalgia was identified as closed pulp teeth with deep caries (36%), followed by vital pulp treatment (29%), and pulpitis or apical periodontitis (14%). This ranking of prevalence is consistent with the present study, which showed that deep fillings (64%) and canal treatments-apical periodontitis (48%) were common.

In a study involving 11,617 personnel in simulated high-altitude flights up to 43,000 ft, only 30 (0.26%) individuals reported toothaches (barodontalgia). Among these individuals, 25 experienced 28 episodes of pain; chronic pulpitis was suspected in 22 cases and maxillary sinusitis was suspected in two cases. No pathologies were identified in four cases. In 10 cases where pulpitis was treated by root filling or the replacement of a deep filling, subsequent low-pressure exposure did not result in pain (*Kollmann, 1993*).

Toothaches at altitudes >5,000 ft can affect both healthy and restored teeth; they are presumed to result from severe pressure changes. The incidence of such pain has persisted despite advancements in flight conditions. These toothaches pose challenges for flyers, including commercial passengers and civilian and military pilots. The occurrence of barodontalgia is influenced by various factors, with potential links between symptoms and underlying pulp pathology. Pathologies may include increased exudate leading to necrosis and hypothermia. Indirect toothaches can arise from upper alveolar nerve stimulation due to maxillary barosinusitis or anatomical malformations. Pain frequently occurs while gaining altitude, and upper molars are most affected. Comprehensive clinical and radiological examinations are crucial. Regular dental checks are important for flight crews, with particular emphasis on preventive treatments. Barodontalgia can be significantly reduced through proper root and tooth care (*Alwohaibi et al., 2020*).

Resin cements are favored in dentistry for the cementing of fixed prostheses in individuals with barodontalgia risk and in members of the general population. These cements retain their bonding strength under pressure gradients and show minimal microleakage compared with other cements. Good oral health is needed to prevent barodontalgia. Clinicians must carefully consider factors such as exposed dentin, deep caries, fractured cusps, restoration integrity, and periapical pathology in at-risk individuals. The Fédération Dentaire Internationale (FDA) has classified barodontalgia into four categories based on symptoms; it has provided valuable specific recommendations for therapeutic intervention (*Ferjentsik & Aker, 1982*; *Al-Hajri & Ebtissam, 2006*; *Zadik, 2009a*).

Changes in ambient pressure during flight, diving, or hyperbaric oxygen therapy can lead to various pathophysiological conditions, including barodontalgia. One case study highlighted a patient with severe pain in the mandibular left first molar area during air travel, which persisted after flight. Despite conservative restoration, radiological examination revealed radiolucency in the periapical region of the distal root. Pain relief was only achieved by endodontic treatment. This case underscores the need for dentists to advise patients against exposure to pressure changes until all necessary surgical, conservative,

and prosthetic procedures are complete, considering the effects of pressure deviation on various pathologies (*Stoetzer et al., 2012*) .

Ambient pressure changes during flight, diving, or hyperbaric oxygen treatment can result in barotrauma. Commercial flight crew members experience minor but prolonged pressure changes, whereas military and acrobatic pilots encounter rapid pressure changes and strong acceleration forces, such as forces approaching 9G in sudden maneuvers. Due to high environmental density, divers are subjected to very high ambient pressures, although these are generally encountered for shorter durations. Barodontalgia occurrence has been reported at flight altitudes of 600–1,500 m and diving depths of 10–25 m (*Kieser & Holborow, 1997*).

Ferjentsik and Aker's widely recognized classification of barodontalgia, established in 1982, is based on underlying causes and clinical symptoms (*Ferjentsik & Aker, 1982*). Barotrauma, defined as damage from pressure changes, is closely related to Boyle's Law, which states that the volume of an ideal gas at constant temperature inversely varies according to pressure. Increased pressure decreases gas volume, whereas decreased pressure increases gas volume. Pain during ascent may indicate serious pulp infection (pulpitis); pain during descent suggests pulp necrosis or facial barotrauma (*Kieser & Holborow, 1997*).

Pressure differences in gas-filled body cavities, not equalized with the external environment, can cause pain, edema, or vascular gas embolism (*Felkai et al., 2023*). This pain typically occurs in the lungs, middle ear (barotitis media), or maxillary sinus (barosinusitis) (*Robichaud & McNally, 2005*; *Kieser & Holborow, 1997*). Barosinusitis often develops in the context of acute or chronic maxillary sinusitis, potentially causing pain, numbness, or toothache in the maxillary posterior region. In the clinical setting, it is particularly challenging to distinguish between barosinusitis and barodontalgia based on maxillary pain (*Kollmann, 1993*; *Jamil, Reilly & Cooper, 2024*).

Barodontalgia is generally defined as dental pain that occurs after pressure changes (*Zadik, Chapnick & Goldstein, 2007*). Initially observed in flight crews during World War II and referred to as ''aerodontalgia'' due to its association with air, the term was later changed to ''barodontalgia'' upon recognition of its occurrence in divers, thereby emphasizing its relation to pressure changes. The incidence of this type of toothache ranges from 0.26% to 2.8% in flight crews, airline passengers, and divers (*Kollmann, 1993*; *Zadik, Chapnick & Goldstein, 2007*; *Taylor, O'Toole & Ryan, 2003*); no statistically significant difference has been noted between divers and flight crews.

In one case study, a 26-year-old man presented to a dental clinic with a 5-day history of pain in the left mandibular region. The pain, which developed suddenly at the end of an airplane's ascent, was rated as 8 on a numerical rating scale from 0 (no pain) to 10 (worst pain possible) (*Stoetzer et al., 2012*). The patient experienced temporary absence of pain for ∼5 h after landing, followed by resurgence of dull, throbbing pain, rated between 6 and 7. In the absence of urgent dental care, the patient managed his pain with 1,600–2,400 mg of ibuprofen daily for 4 days. Clinical examination revealed that the mandibular left first molar (tooth no: 36), which had undergone conservative restoration, was sensitive to percussion. No periodontal pathologies were detected in the left mandible, and a vitality test indicated the tooth was devitalized (*Stoetzer et al., 2012*).

The exact etiology of barodontalgia remains unclear; thus, current dental treatment recommendations for flight and diving personnel are based on statistical data (*Goethe, Bater & Laban, 1989*). Barodontalgia can develop regardless of the type of pressure change, and it may persist despite pressure equalization (*Gonzalez Santiago, Martinez-Sahuquillo Marquez & Bullon-Fernandez, 2004*). Potential etiologies include dental infections, variations in the expansion levels of tooth enamel and pulp against pressure, and pressure-induced fluid movements from exposed dentin to the pulp (*Jamil, Reilly & Cooper, 2024*; *Carlson, Halverson & Triplett, 1983*; *Zadik et al., 2006*). Animal experiments have shown that fluid can pass from dentin to the pulp chamber under hyperbaric conditions after cavity preparation in enamel (*Carlson, Halverson & Triplett, 1983*). Retrospective studies have identified caries lesions extending to the dentin or faulty restorations in most patients with distinct barodontalgia (*Zadik & Einy, 2006*). From a clinical perspective, this finding implies that patients with caries lesions or patients undergoing dental treatments that expose dentin, such as prosthetic tooth preparation, should avoid exposure to pressure changes until definitive treatment is completed.

In the literature, periapical inflammation and pulpitis after dental restoration have been identified as prevalent causes of barodontalgia. The case presented here involved pain in a tooth with pulp necrosis; the patient remained asymptomatic until exposure to pressure changes during flight. Despite a lack of exposure to further pressure changes in a hyperbaric environment, the pain developed and persisted. The pathophysiology of this pain is not fully understood, but it is presumed to involve disrupted micro-circulation in the pulp.

In addition to toothache resulting from pressure changes in pilots and divers, there is evidence that such changes can lead to fractures in teeth or dental restorations (*Gunepin, Derache & Audoual, 2010*; *Calder & Ramsey, 1983*; *Woodmansey, 2008*). However, in the case described by *Calder & Ramsey (1983)*, no such fractures were observed. Although it was once believed that trapped gas between a tooth and its restoration led to these fractures and subsequent barodontalgia, the results of recent studies suggest that secondary caries constitute the underlying cause. Today, it is believed that fluid movement in carious dentin from the tubules to the pulp can cause pulpitis (*Calder & Ramsey, 1983*; *Zadik, 2010*).

Indirect pulp capping is contraindicated for patients subject to pressure changes; these changes are presumed to negatively impact pulp tissue regeneration. To prevent potential complications, endodontic treatment or—in more severe cases—tooth extraction should be considered when direct pulp crowning is indicated. Comprehensive dental examinations are essential before exposure to pressure changes. Treatments should address all caries lesions, faulty restorations, and existing inflammation. Vitality tests should be conducted on all teeth to detect and treat asymptomatic pulp necrosis (*Sumen, Dumlu & Altun, 2023*; *Zadik, 2009b*; *Felkai et al., 2023*). Overall, it is reasonable to prohibit pilots and flight crew from flying until the completion of treatments for periodontitis or apical periodontitis, due to the risk of barodontalgia.

## Study limitations

One limitation of the present study is high number of pilots refusing to participate in the study. Although 750 pilots were asked to fill in the questionnaire, 224 of them refused to

participate. Those with refusal might be less interested in the topic or have not been affected by barodontalgia. These might introduce self-selection bias, which makes the barodontalgia prevalence look higher than the actual prevalence. Additionally, since this survey is a study that is not administered directly to the participant, it may lead to additional bias.

## CONCLUSION

Through this study, we concluded that dental and orofacial pain were experienced by more than half of the pilots at least once during their flight. It was observed that awareness was higher among pilots experiencing barodontalgia, especially jet pilots, and they went to the dentist more often. The results suggest a significant impact of barodontalgia on pilot performance and flight safety, even at lower altitudes. Thus, there is a need to educate pilots about stress management, barodontalgia awareness, and the importance of regular dental check-ups.

### Funding
The authors received no funding for this work.

### Competing Interests
The authors declare there are no competing interests.

### Author Contributions

- Celalettin Topbaş conceived and designed the experiments, performed the experiments, prepared figures and/or tables, authored or reviewed drafts of the article, and approved the final draft.
- Dursun Ali Şirin conceived and designed the experiments, authored or reviewed drafts of the article, and approved the final draft.
- Hilal Gezeravcı conceived and designed the experiments, performed the experiments, prepared figures and/or tables, and approved the final draft.
- Fatih Özçelik analyzed the data, prepared figures and/or tables, authored or reviewed drafts of the article, and approved the final draft.
- Yelda Erdem Hepşenoğlu analyzed the data, authored or reviewed drafts of the article, and approved the final draft.
- Şeyda Erşahan analyzed the data, authored or reviewed drafts of the article, and approved the final draft.

### Human Ethics

The following information was supplied relating to ethical approvals (*i.e.,* approving body and any reference numbers):

The research underwent evaluation by the Hamidiye Scientific Research Ethical Board at the University of Health Sciences meeting on November 4, 2022. It received ethical approval with decision number 24/15.

## Data Availability

The raw data is available in the Supplemental File.

## Supplemental Information

Supplemental information for this article can be found online at http://dx.doi.org/10.7717/peerj.17290#supplemental-information.

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
