# Peer review of "Relationships among barodontalgia prevalence, altitude, stress, dental care frequency, and barodontalgia awareness: a survey of Turkish pilots"

_PeerJ, doi:10.7717/peerj.17290_

## Round 0.1 · original submission · Major Revisions

The article must be reviewed, particularly the statistical analysis section. Update bibliographic references.

Reviewer 1 ·

Basic reporting

English is clear.
Unfortunately, many references are outdated, from the 1980s and 1990s. They need to be removed. Additionally, there is a lack of recent publications that should be addressed and updated.

Experimental design

In the study, a questionnaire was utilized. A significant advantage of the paper is the large number of respondents, 526. Despite being straightforward, the research yields meaningful results.

Validity of the findings

The study involves civilian and military pilots investigating the relationship between barodontalgia, frequency of dental visits, and the prevalence of such pain among pilots operating at high altitudes. Out of 750 surveyed, 526 completed the study. 61% of pilots were aware of barodontalgia, and 81% of them experienced pain below 2000 feet. Higher awareness was found among pilots who had experienced barodontalgia, especially among jet pilots. Those with barodontalgia had more frequent dental visits. The results suggest a significant impact of barodontalgia on pilot performance and flight safety, even at lower altitudes.

Additional comments

The work is well-prepared and worthy of publication.

Reviewer 2 ·

Basic reporting

In line 94, please elaborate on what descriptive statistics were used. In lines 94-96, please describe what variables were compared between what groups by using either chi-square test or m-w u test. Please also indicate how a statistical significance was determined in your study. For example, “two-sided p-values < 0.05 indicate statistical significance.”

Experimental design

A total of 526 out of 750 pilots agreed to participate in the study. Those pilots who refused to participate might be less interested in the topic or have not been affected by barodontalgia. These might introduce self-selection bias, which makes the barodontalgia prevalence look higher than the actual prevalence. Could you add a few sentences to discuss this as one of your study limitations?

Validity of the findings

no comment

·

Basic reporting

I appreciate the opportunity to have reviewed this manuscript on the prevalence of barodontalgia in a pilot population in Turkey. The study is interesting and provides more information about the prevalence and impact of barodontalgia in a certain population.
The chapters of the manuscript are well structured, but I would recommend a review of the English language.
The authors used terms that do not match the translation. For example: tartar cleaning.
As I understand it, the tables should be open laterally. When it doesn't, it's a chart. In addition, all tables were named table 1.
The following reference could have been used in the text: Prevalence of barodontalgia in Brazilian aviation pilots and flight attendants. DOI: 10.25259/IJASM_21_2022

Experimental design

In my opinion, the study design needs to be revised.
How was the randomization performed in this study?
If the study was based on the application of a questionnaire sent by e-mail, how can it be considered a clinical study?
How was the blinding done? At what point in the study did this happen?
Statistical analysis needs to be written down in more detail.

Validity of the findings

The discussion chapter needs to be rewritten. I found the chapter to be very long and should be focused on each result of the study. Authors need to discuss the results and not just present the results and compare them with findings in the literature.
Take the following example:
In the second paragraph: The study revealed thar 45% of the participants did not experienced any in-flight pain; 29.5%, 17.4%, and 8% encountered barodontalgia once, twice and >3 times, respectively. What would explain these findings? What might have contributed?
In the text (fourth paragraph), there is a quotation from Kennebeck et al., 1946. There is no other more recent study that talks about this information.
What are the limitations of the study?
As this is a study based on a questionnaire that was not applied directly to the participant, this can generate a number of biases. The authors could discuss this possibility.
I disagree with the conclusions. They should be focused on the results of the study. The authors could use this text of the conclusion to finalize the chapter of the discussion and draft a new one.

---

## Round 0.2 · accepted · Accept

The article is now ready to be published. Please review the small changes suggested.

Reviewer 2 ·

Basic reporting

The authors have sufficiently addressed my comments. I have no further comments.

Experimental design

The authors have sufficiently addressed my comments. I have no further comments.

Validity of the findings

The authors have sufficiently addressed my comments. I have no further comments.

·

Basic reporting

No comment.

Experimental design

In the line 9 of the chapter Methods, the authors say…The minimum sample size required for this study, with α=0.05 and power=0.80, was initially calculated as 14 subjects per group. However, there is only one group in the study. Therefore, this sentence needs to be corrected.
At the end of the description of the statistical analysis, the authors should mention the level of significance considered in the analysis.

Validity of the findings

In the chapter results:
In the sentence “The majority of affected pilots reported that pain mostly occurred during descent, you should write “Most of affected pilots reported that pain mostly occurred during descent.”

Additional comments

I could observe that there was an improvement in manuscript comprehension, especially in relation to the English language. However, I still suggest minor revisions.